# Sliding Mode Disturbance Observer-Based Adaptive Dynamic Inversion Fault-Tolerant Control for Fixed-Wing UAV

**Zhe Dong** [1], **Kai Liu** [1,*] and **Shipeng Wang** [2]

1 School of Aeronautics and Astronautics, Dalian University of Technology, Dalian 116024, China
2 Shenyang Aircraft Design and Research Institute, Shenyang 110034, China
* Correspondence: carsonliu@dlut.edu.cn

**Abstract:** Unmanned aerial vehicles (UAVs) have been widely applied over the past decades, especially in the military field. Due to the unpredictability of the flight environment and failures, higher requirements are placed on the design of the control system of the fixed-wing UAV. In this study, a sliding mode disturbance observer-based (SMDO) adaptive dynamic inversion fault-tolerant controller was designed, which includes an outer-loop sliding mode observer-based disturbance suppression dynamic inversion controller and an inner-loop real-time aerodynamic identification-based adaptive fault-tolerant dynamic inversion controller. The sliding mode disturbance observer in the outer-loop controller was designed based on the second-order super-twisting algorithm to alleviate chattering. The aerodynamic identification in the inner-loop controller adopts the recursive least squares algorithm to update the aerodynamic model of the UAV online, thereby realizing the fault-tolerant control for the control surface damage. The effectiveness of the proposed SMDO enhanced adaptive fault-tolerant control method was validated by mathematical simulation.

**Keywords:** unmanned aerial vehicle; disturbance suppression control; aerodynamic identification; recursive least squares; super-twisting algorithm; chattering reduction; control surface damage



## 1. Introduction

Unmanned aerial vehicles (UAVs) are currently playing an increasingly important role in various fields [1–3]. In particular, unmanned combat aircraft have become important for strategic command in air combat, emphasizing high reliability and strong adaptability [4]. Compared with manned fighter aircraft, unmanned aircraft overcome the limitations of the pilot's physiological conditions, and thus have great potential to increase the load. This creates significant room for the improvement in the combat effectiveness of unmanned combat aircraft [5].

Due to the above characteristics and working environment of UAVs, fault-tolerant control and disturbance suppression are particularly important [6]. Therefore, studying the fault-tolerant control method of UAVs is necessary to improve its reliability and safety, reduce potential safety hazards, and prevent catastrophic accidents [7]. In addition, considering that the dynamics of UAVs are nonlinear and complex, and there are uncertainties in system dynamics and disturbances, it is crucial to develop a robust controller that can effectively suppress disturbances [8].

Despite technological advancement in UAVs, failures are inevitable. This is mainly due to the fact that the UAV is embedded with various subsystems that are susceptible to failures. Furthermore, unforeseen situations and events may occur in their work environment. This reality places new demands on the design and application of fault-tolerant control. Fault-tolerant control is an effective method to improve the robust and reliable operation of UAV. It contains different complex control algorithms that provide possible solutions for fault compensation and control of the system with acceptable performance [9]. Fault-tolerant control can be divided into passive fault tolerance and active fault tolerance. Passive

fault-tolerant control does not rely on fault information for control, and is closely related to robust control, where a fixed controller is designed to be robust to a predefined fault in the system [10]. Active fault control uses a fault detection module to detect and isolate faults, while a supervisory controller decides how to modify the control structure and parameters to compensate for faults that occurred in the system [10]. In terms of passive fault-tolerant control, ref. [11] proposed a piecewise linear assumption, which allows the fault-tolerant control problem to be transformed into a nonlinear control allocation problem. The method was applied to control a solar-powered UAV with control effector faults. Ref. [12] presented an adaptive attitude finite time tracking control algorithm for a quadrotor unmanned aerial vehicle in the presence of actuator faults, which is based on the non-singular terminal sliding mode algorithm. In terms of active fault-tolerant control, ref. [13] designed a gain schedule-based fault-tolerant control approach in the framework of structured H∞ synthesis for aUAV with actuator faults. The effectiveness of the method was experimentally demonstrated on a hexacopter UAV. Ref. [14] developed an active fault-tolerant controller for the attitude control system of a fixed-wing UAV with control surface failures and external disturbances, which includes a neural network-based fault estimation observer for fault detection. In [15], an active fault-tolerant control method for actuator failures of a the fixed-wing UAV was proposed. Nonlinear dynamic inversion was combined with a neural network adaptive module for actuator failure regulation.

In the field of UAV control, the nonlinear dynamic inversion control method has been widely applied, which provides a compromise between controller complexity and performance [16]. However, the single dynamic inversion control method has the disadvantage of being sensitive to model errors and external disturbances [17]. One solution is to design a composite control structure including a baseline NDI controller and a nonlinear disturbance observer to enhance the robustness of the closed-loop system. The basic idea of a disturbance observer is to treat all internal uncertainties, external disturbances, parameter changes and unmodeled dynamics as lumped disturbances [18]. In recent years, nonlinear disturbance observer [19] and the time-delay estimation [20] method have been widely used, but they require the state and its derivative information to achieve high-precision disturbance estimation performance. To avoid the above problem, an extended state observer was developed to estimate the lumped disturbance and system states [21]. However, the high order of the extended state observer causes computational burden and slows thetransient response. One solution to this difficulty is the sliding mode observer [22]. In practical applications, in order to suppress the chattering of the sliding mode observer, high-order sliding mode observers have been widely used. Among these observers, the super-twisting algorithm-based second-order sliding mode observer can suppress the chattering while avoiding the complex high-order derivation [23].

To summarize, many linear and nonlinear control approaches have been studied for the fault-tolerant control problem of UAVs. However, most research did not consider the influence of aerodynamic model changes on control after UAV failure. In addition, the combined application of the disturbance observer and fault-tolerant control should be fully considered to achieve better reconfigurable control performance.

This paper focuses on the attitude control problem of the fixed-wing UAV under the influence of disturbances and faults. First, the three-channel dynamic model of the UAV is established, including the attitude angle dynamic equations and the rotational dynamic equations. Control surface failure factors are introduced into the aerodynamic moment to simulate the partial loss of control surfaces. Then, a sliding mode disturbance observer-based nonlinear dynamic inversion (SMDO-NDI) controller is designed. Taking the nonlinear dynamic inversion as the baseline controller, a super-twisting algorithm-based sliding mode disturbance observer is introduced to suppress the inherent sensitivity of the dynamic inversion method to model errors and external disturbances. Finally, an adaptive disturbance suppression integrated controller (ADSIC) is designed. The controller integrates the designed sliding mode disturbance observer into the NDI outer control loop, which is more susceptible to disturbances, while the NDI inner loop adopts a real-time

aerodynamic identification-based adaptive nonlinear dynamic inversion (ANDI) control method for fault-tolerant control of actuators, and updates the aerodynamic model online to realize the reconfigurable control.

## 2. Mathematical Modeling of the Fixed-Wing UAV under Control Surface Damage

The starting point for the attitude controller design is the equations of motion describing the fixed-wing UAV rigid-body rotation. The dynamic and kinematic equations of the UAV rotational motion are as follows [24,25]:

$$
\begin{cases}
\dot{\phi} = p + \tan\theta(q\sin\phi + r\cos\phi) \\
\dot{\theta} = q\cos\phi - r\sin\phi \\
\dot{\beta} = p\sin\alpha - r\cos\alpha + \frac{1}{mV}(Y - F_T\cos\alpha\sin\beta + mg_2) \\
\dot{p} = \dfrac{I_z(\overline{L}_o + \overline{L}_{\delta_a}) + I_{xz}\overline{N}}{I_x I_z - I_{xz}^2} \\
\dot{q} = \dfrac{(\overline{M}_o + \overline{M}_{\delta_e})}{I_y}v \\
\dot{r} = \dfrac{I_x(\overline{N}_o + \overline{N}_{\delta_r}) + I_{xz}\overline{L}}{I_x I_z - I_{xz}^2}
\end{cases}
\tag{1}
$$

where $\phi$ is the roll angle, $\theta$ the pitch angle, and $\beta$ the sideslip angle. $p$, $q$, and $r$ are the roll, pitch, and yaw angular rates, respectively. Angular rates are assumed to be the fast states because the control surface deflections aileron $\delta_a$, elevator $\delta_e$, and rudder $\delta_r$ have a significant, direct effect on $\dot{p}$, $\dot{q}$, and $\dot{r}$ [26]. $Y$ is the side force expressed in the wind-axes reference frame and $F_T$ is the thrust force. $\overline{L}$, $\overline{M}$, and $\overline{N}$ are the roll, pitch and yaw moment, respectively. Among them, the subscript $o$ represents the moment about the vehicle center of mass, and the subscript $\delta_{a,e,r}$ represents the external moment due to the control surface deflections. The aerodynamic models of $\overline{L}_{\delta_a}$, $\overline{M}_{\delta_e}$, $\overline{N}_{\delta_r}$, and the gravity component $g_2$ are defined and formulated as the following:

$$
\begin{cases}
\overline{L}_{\delta_a} = \overline{q}Sb\left(C_{l0}(\alpha,\beta,p,r) + f_a C_l^{\delta_a}\delta_a\right) \\
\overline{M}_{\delta_e} = \overline{q}S\overline{c}\left(C_{m0}(\alpha,q) + f_e C_m^{\delta_e}\delta_e\right) \\
\overline{N}_{\delta_r} = \overline{q}Sb\left(C_{n0}(\alpha,\beta,p,r) + f_r C_n^{\delta_r}\delta_r\right)
\end{cases}
\tag{2}
$$

$$
g_2 = g(\cos\alpha\sin\beta\sin\theta + \cos\beta\sin\phi\cos\theta - \sin\alpha\sin\beta\cos\phi\cos\theta),
\tag{3}
$$

where $\overline{q}$ is the dynamic pressure, $b$ the reference wing span, $\overline{c}$ the mean aerodynamic chord and $S$ the reference wing surface area. $C_{l0}(\alpha,\beta,p,r)$ is the component of the roll moment coefficient other than the aileron efficiency term, $C_{m0}(\alpha,q)$ the component of the pitch moment coefficient other than the elevator efficiency term and $C_{n0}(\alpha,\beta,p,r)$ the component of the yaw moment coefficient other than the rudder efficiency term. $f_a$, $f_e$ and $f_r$ are the efficiency attenuation coefficients of the control surface under the structural damage of the aileron, elevator, and rudder, respectively.

According to the principle of singular perturbation, the state variables can be divided into two loops with different speeds, the attitude angle loop and the angular rate loop, for the control law design [27]. The attitude angles $\phi$, $\theta$ and $\beta$ form the outer loop, and the angular rates $p$, $q$, and $r$ form the inner loop. Let $x_1 = \begin{bmatrix} \phi & \theta & \beta \end{bmatrix}^T$, $x_2 = \begin{bmatrix} p & q & r \end{bmatrix}^T$; then, the state space form of the nonlinear dynamics model of the UAV can be written as [28]:

$$
\begin{cases}
\dot{x}_1 = f_s + g_s\dot{x}_2 + D_s \\
\dot{x}_2 = f_f + g_f u + D_f \\
y = x_1
\end{cases}
\tag{4}
$$

where $D_s = \Delta f_s + \Delta g_s \dot{x}_2 + d_s$ and $D_f = \Delta f_f + \Delta g_f u + d_f$ represents the composite disturbance of the system. $\Delta f_s$ and $\Delta g_s$ are the modeling error and internal uncertainty of the outer loop of the system, respectively. $\Delta f_f$ and $\Delta g_f$ are the modeling error and internal uncertainty of the inner loop of the system, respectively. $d_s$ and $d_f$ are the external disturbances of the two loops.

## 3. Sliding Mode Disturbance Observer-Based Dynamic Inversion Controller Design

In this section, a SMDO-NDI control law is designed. As a typical feedback linearization control method, dynamic inversion can realize the control of UAV with nonlinear characteristics. However, the dynamic inversion control method is sensitive to model errors and external disturbances. Therefore, it is considered to introduce a sliding mode disturbance observer to suppress errors and disturbances, reduce the dependence of the control method on the accurate model, and improve the robustness of the control system.

The structure flow diagram of the sliding mode disturbance observer-based dynamic inversion controller is as follows:

As shown in Figure 1, the error signal $e$ formed by the state output $y$ and its nominal value $y_c$ passes through the baseline nonlinear dynamic inversion controller and outputs the control signal $u_{\delta,n}$. The sliding mode disturbance observer integrates the control input $u$ and the state variable $x$, and estimates the composite disturbance to obtain $\hat{D}$, thereby forming the compensation control signal $u_{\delta,o}$, which together with $u_{\delta,n}$ constitutes the final control signal $u$.

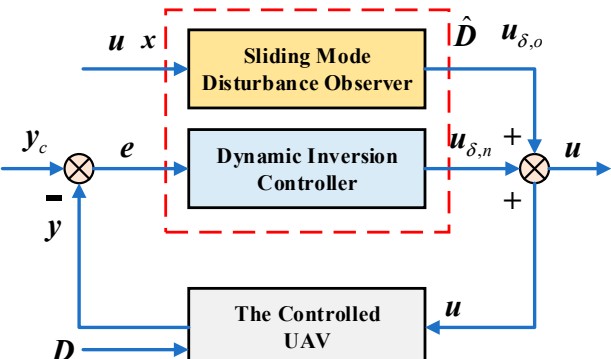

**Figure 1.** Sliding mode disturbance observer-based nonlinear dynamic inversion (SMDO-NDI) control structure flow diagram.

### 3.1. Disturbance Compensation Dynamic Inversion Controller

First, a general disturbance compensation dynamic inversion controller is designed according to the general form of multi-input multi-output nonlinear system. Let the MIMO affine nonlinear uncertain system be as follows:

$$\begin{cases} \dot{x} = f(x,t) + \Delta f(x,t) + (g(x,t) + \Delta g(x,t))u(x,t) + d(t) \\ y = x \end{cases}. \tag{5}$$

The variable definitions in the above formula are the same as those in the previous section. Write $\Delta f$, $\Delta g$ and $d$ as a term $D$, that is, $D = \Delta f + \Delta g + d$, then system Equation (5) can be simplified as

$$\begin{cases} \dot{x} = f(x,t) + g(x,t)u(x,t) + D(x,t) \\ y = x \end{cases}. \tag{6}$$

In the dynamic inversion control, the virtual control variable $v$ is often introduced to make a linear relationship between the one-step differential of the state variable and the virtual control variable; then, the dynamic inversion control law of the nonlinear system Equation (6) affected by disturbance can be written a:

$$\boldsymbol{u}_\delta = \boldsymbol{g}^{-1}(-\boldsymbol{f} + \boldsymbol{v} - \boldsymbol{D}). \tag{7}$$

Let $\boldsymbol{v} = \boldsymbol{\omega}_d(\boldsymbol{x}_c - \boldsymbol{x})$., where $\boldsymbol{\omega}_d$ is the bandwidth of the controlled system [29]; then, the nonlinear system is converted into a first-order multivariable linear decoupling system.

In the control law Equation (7), the composite disturbance $\boldsymbol{D}$ is an unknown quantity, so the control quantity $\boldsymbol{u}_\delta$ is an ideal dynamic inversion control law, which cannot be applied in practical situations. If $\boldsymbol{D}$ is ignored, then the above ideal control law is converted to a nominal control law:

$$\boldsymbol{u}_{\delta,n} = \boldsymbol{g}^{-1}(-\boldsymbol{f} + \boldsymbol{v}). \tag{8}$$

However, if only the nominal control law is used, when the influence of the composite disturbance $\boldsymbol{D}$ cannot be ignored, the performance of the dynamic inversion control law cannot be guaranteed, resulting in the instability of the controlled system, and even the phenomenon of divergence. Therefore, it is necessary to add control compensation to the composite disturbance in the nominal control law.

Sliding mode control is invariant to uncertainty and disturbance, and the sliding mode disturbance observer has the advantages of easy implementation and fast convergence [30]; thus the sliding mode disturbance observer is introduced to estimate the composite disturbance $\boldsymbol{D}$ and compensate the nominal dynamic inversion control law. Assuming that the estimated value of the sliding mode observer for the composite disturbance is $\hat{\boldsymbol{D}}$, the disturbance compensation control law is designed according to the nonlinear uncertain system Equation (6) as:

$$\boldsymbol{u}_{\delta,o} = -\boldsymbol{g}^{-1}\hat{\boldsymbol{D}}. \tag{9}$$

Combining the nominal control law Equation (8) and the disturbance compensation control law Equation (9) together, the disturbance compensation dynamic inversion control law based on the sliding mode disturbance observer is obtained:

$$\boldsymbol{u} = \boldsymbol{u}_{\delta,n} + \boldsymbol{u}_{\delta,o}. \tag{10}$$

*3.2. Super-Twisting Algorithm-Based Sliding Mode Disturbance Observer*

Aiming at the shortcoming that the dynamic inversion control method is sensitive to model errors and external disturbances, the sliding mode disturbance observer is used to compensate for the dynamic inversion in this paper.

In this section, a sliding mode disturbance observer based on the super-twisting algorithm is used to estimate the disturbance. The super-twisting algorithm is a high-order sliding mode control algorithm, which can realize the stable convergence of the sliding mode variable and its first derivative to 0 in a finite time for bounded disturbances. At the same time, since the high-frequency switching part of the algorithm is hidden in the high-order derivative of the sliding mode variable, chattering can be effectively suppressed.

Two assumptions are posed before designing the super-twisting algorithm-based sliding mode observer [31]:

**Assumption 1.** *All states of the controlled system are observable.*

**Assumption 2.** *The partial derivative of the composite disturbance $\boldsymbol{D}$ with respect to time is continuous and bounded, that is, there is a known bounded constant $C > 0$ that makes $\sup\limits_{t \in [0,\infty)} \left| \frac{\partial \boldsymbol{D}}{\partial t} \right| \leqslant C$ true.*

The super-twisting algorithm is obtained [32] based on the analysis of the perturbed non-linear differential equation:

$$\dot{x}(t) + w_1 |x(t)|^{1/2} \mathrm{sgn}x(t) + w_2 \int \mathrm{sgn}x(\tau) d\tau = \zeta(t), \tag{11}$$

where $\zeta(t)$ is the unknown bounded disturbance, and $\left| \dot{\zeta}(t) \right| \leq Q$, $Q$ is the upper bound of the derivative of the disturbance, and $w_1$, $w_2$ are constant coefficients. It is well known [33]

that a solution $x(t)$ of Equation (11) and its derivative $\dot{x}(t)$ converge to 0 in finite time $t_{con} \leqslant 7.6x(0)/(w_2 - Q)$ if $w_1 \geqslant 1.5\sqrt{Q}$ and $w_2 \geqslant 1.1Q$.

For the nonlinear uncertain system Equation (6), the sliding mode disturbance observer can be constructed as [34]

$$\begin{cases} \boldsymbol{s} = \boldsymbol{x} - \boldsymbol{r} \\ \dot{\boldsymbol{r}} = \boldsymbol{f} + \boldsymbol{gu} + \hat{\boldsymbol{D}} \\ \hat{\boldsymbol{D}} = \boldsymbol{w}_1 |\boldsymbol{s}|^{1/2}\text{sgn}\boldsymbol{s} + \boldsymbol{w}_2 \int \text{sgn}\boldsymbol{s}d\tau \end{cases}, \tag{12}$$

where $\boldsymbol{s}$ is the auxiliary sliding mode surface, and $\hat{\boldsymbol{D}}$ is the observed value of the composite disturbance. $\boldsymbol{w}_1$ and $\boldsymbol{w}_2$ in $\hat{\boldsymbol{D}}$ are diagonal matrices composed of constant coefficients. It is specified that $|\boldsymbol{s}|^{1/2}\text{sgn}\boldsymbol{s}$ in Equation (12) is calculated as follows:

$$|\boldsymbol{s}|^{1/2}\text{sgn}\boldsymbol{s} = \begin{bmatrix} |s_1|_{1/2}\text{sgn}s_1 \\ |s_2|_{1/2}\text{sgn}s_2 \\ \vdots \\ |s_n|_{1/2}\text{sgn}s_n \end{bmatrix}. \tag{13}$$

Taking the derivative with respect to $\boldsymbol{s}$ and according to the system Equation (6), we obtain:

$$\dot{\boldsymbol{s}} = \dot{\boldsymbol{x}} - \dot{\boldsymbol{r}} = \boldsymbol{f} + \boldsymbol{gu} + \boldsymbol{D} - \boldsymbol{f} - \boldsymbol{gu} - \hat{\boldsymbol{D}} = \boldsymbol{D} - \hat{\boldsymbol{D}}. \tag{14}$$

It can be shown from Equation (14) that $\hat{\boldsymbol{D}}$ converge to $\boldsymbol{D}$ in finite time, that is, the composite disturbance observed value converges to its true value in a finite time.

After the theoretical derivation of the dynamic inversion control law and the sliding mode observer is completed, a specific adaptive disturbance suppression integrated controller is designed for the UAV.

## 4. Adaptive Disturbance Suppression Integrated Controller Design for UAV

This section designs the controller of the UAV for the specific problem. A sliding mode disturbance observer is considered to compensate the outer loop of the baseline dynamic inversion controller, which often faces greater disturbances. When the control surface of the UAV has structural damage, such as the lack of a control surface, its aerodynamic model will change; in particular, especially the efficiency of the control surface will decrease significantly. Therefore, a real-time aerodynamic identification module is introduced to adaptively adjust the dynamic inversion inner-loop control moment. The structure flow diagram of the ADSIC is shown in Figure 2.

As can be seen from the figure, for the sliding mode disturbance observer, the input is the outer-loop disturbance estimation $\hat{\boldsymbol{D}}_s$ and control variable $\boldsymbol{u}_1$, namely $\boldsymbol{x}_2$, and the output is the outer-loop compensation control law $\boldsymbol{x}_{2\delta,o}$. For the adaptive module, the aerodynamic identification adaptively revises the inner-loop control law $\boldsymbol{M}_{\delta,c}$, and the error between the estimation of angular rate and its actual value is used to offset uncertainty in the control law.

### 4.1. Sliding Mode Disturbance Observer-Based Outer-Loop Dynamic Inversion Control Law

The attitude control loop of UAV usually faces larger disturbances, so it is considered to compensate the outer loop of the baseline controller by using the sliding mode disturbance observer.

Based on the nonlinear state space model Equation (4) of the UAV, the disturbance compensation control law Equations (8)–(10) and the sliding mode disturbance observer Equation (12) are combined to obtain the sliding mode disturbance observer-based dynamic inversion outer-loop control law:

$$\begin{cases} x_{2c} = x_{2\delta,n} + x_{2\delta,o} \\ \begin{cases} x_{2\delta,n} = g_s^{-1}(-f_s + \omega_{d,1}(x_{1c} - x_1)) \\ x_{2\delta,o} = -g_s^{-1}\hat{D}_s \\ s_1 = x_1 - r_1 \\ \dot{r}_1 = f_s + g_s x_2 + \hat{D}_s \\ \hat{D}_s = w_1|s_1|^{1/2}\text{sgn}s_1 + w_2 \int \text{sgn}s_1 d\tau \end{cases} \end{cases}, \tag{15}$$

where $x_{2c}$ is a vector composed of angular rate, which is used as the control variable of the dynamic inversion control law. A gain matrix $K_s$ is usually introduced in specific calculations, and $x_{2\delta,o}$ is approximately equal to $K_s\hat{D}_s$.

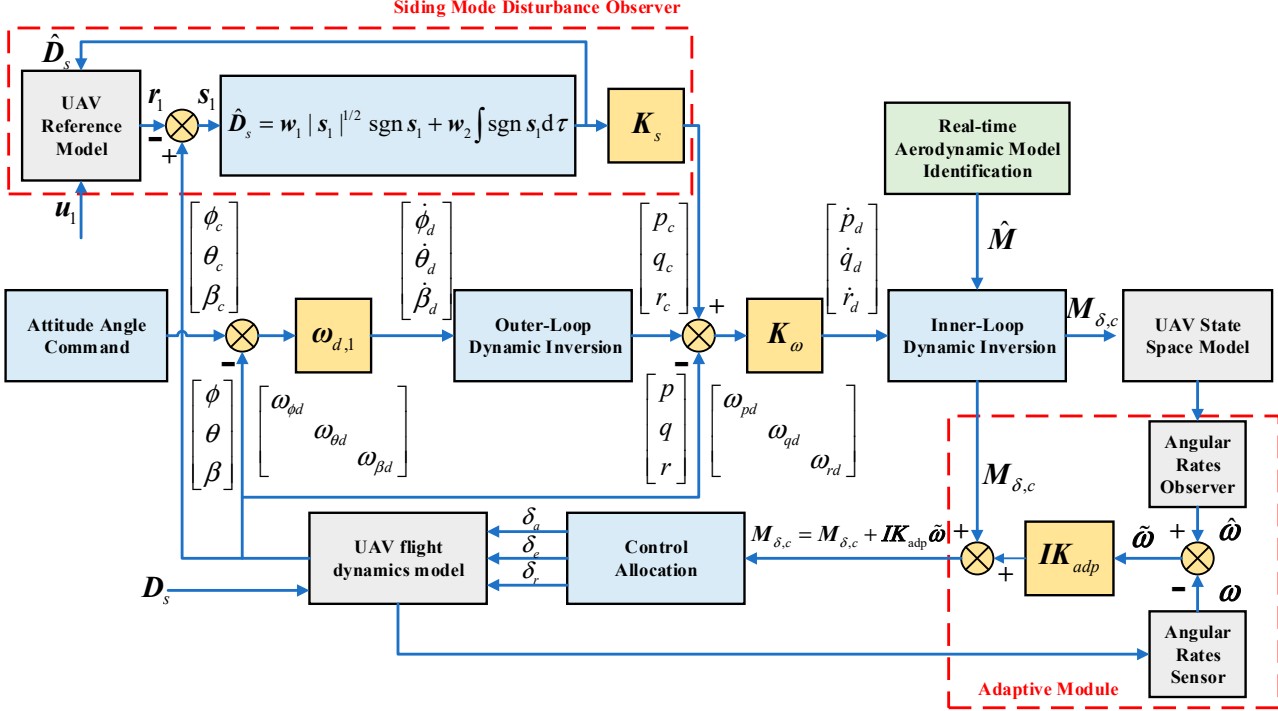

**Figure 2.** Adaptive disturbance suppression integrated controller (ADSIC) control structure flow diagram.

After the design of the dynamic inversion outer-loop disturbance compensation control law is completed, the real-time aerodynamic identification module is further introduced into the inner-loop dynamic inversion control law, and the aerodynamic model is adaptively adjusted to realize fault-tolerant control of control surface faults.

*4.2. Real-Time Aerodynamic Identification-Based Inner Loop Dynamic Inversion Control Law*

4.2.1. Recursive Least Squares-Based Aerodynamic Identification Algorithm

The recursive least squares method [35] is used to achieve an efficient aerodynamic real-time identification task after structural failure of the control surface. The flow diagram of the algorithm is presented in Figure 3.

Taking the aerodynamic moment of the pitch channel as an example, the polynomial aerodynamic model of the pitch moment coefficient is set as follows:

$$C_m = C_{m0} + C_m^\alpha \alpha + C_m^{\delta_e}\delta_e + C_m^q q = H_e\theta_e, \tag{16}$$

where $H_e$ is the identification matrix, $\theta_e$ is the parameter to be identified, and $C_m$ is the state measurement value.

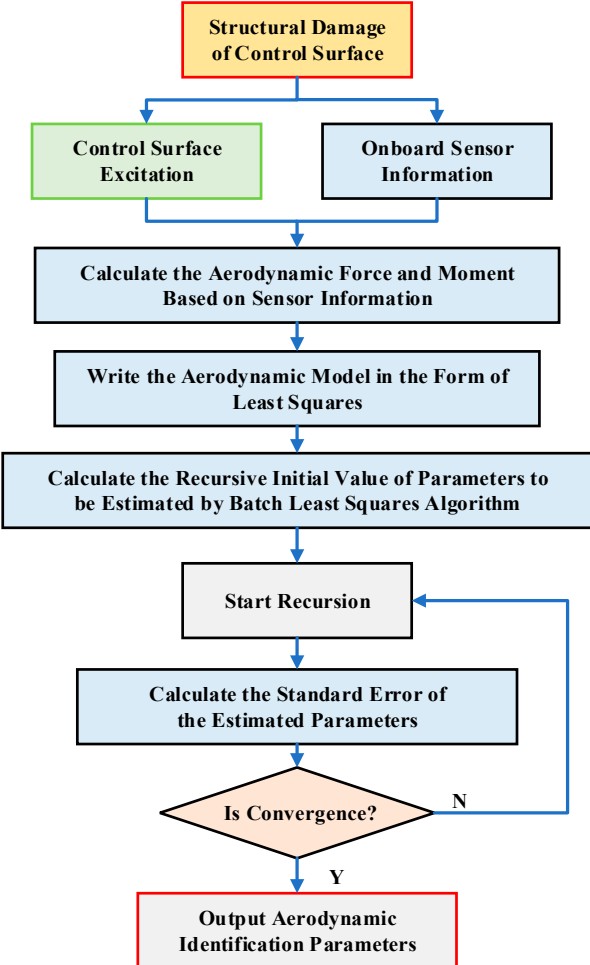

**Figure 3.** Recursive least squares-based aerodynamic identification flow diagram.

With the longitudinal aerodynamic model Equation (16), the aerodynamic parameter vector $\boldsymbol{\theta}_e$ can be estimated according to the real-time measured moment value during the actual flight of the UAV. When the control surface has structural failure, the surface efficiency parameter $C_m^{\delta_e}$ in $\boldsymbol{\theta}_e$ will change, and it is introduced into the control law in the form of an adaptive term to realize the adaptive fault-tolerant control and control allocation of the UAV.

The parameter $\boldsymbol{\theta}_e$ is identified by the recursive least squares identification method, and the recursive formula is as follows [36]:

$$\begin{cases} \boldsymbol{K}(k) = \boldsymbol{P}(k-1)\boldsymbol{H}_e^T(k)\big[1 + \boldsymbol{H}_e(k)\boldsymbol{P}(k-1)\boldsymbol{H}_e^T(k)\big]^{-1} \\ \hat{\boldsymbol{\theta}}_e(k) = \hat{\boldsymbol{\theta}}_e(k-1) + \boldsymbol{K}(k)\big[\boldsymbol{Z}(k) - \boldsymbol{H}_e(k)\hat{\boldsymbol{\theta}}_e(k-1)\big] \\ \boldsymbol{P}(k) = \boldsymbol{P}(k-1) - \boldsymbol{K}(k)\boldsymbol{H}_e(k)\boldsymbol{P}(k-1) \end{cases} , \qquad (17)$$

where $\boldsymbol{K}(k)$ is the parameter iteration scale matrix of the $k$th step, $\boldsymbol{P}(k)$ the identification covariance matrix of the th step, and $\boldsymbol{Z}(k)$ the observation value of the $k$th step.

It is known from the recursive least squares algorithm in Equation (17) that the initial values of the parameter $\hat{\boldsymbol{\theta}}_e$ to be identified and the covariance matrix $\boldsymbol{P}$ need to be given before the recursion starts. The method of assigning initial values here is to collect the state vector $\boldsymbol{H}_e$ and observation value $\boldsymbol{Z}$ for a period of time during the actual flight, and then use batch processing to solve the initial values of $\hat{\boldsymbol{\theta}}_e$ and $\boldsymbol{P}$. Assuming that the sampling interval during this period is $k = 0 \sim N_0$, the batch least squares-based initialization process can be written as follows:

$$\begin{cases} \boldsymbol{B}(N_0) = \boldsymbol{H}_e^{\mathrm{T}}(N_0)\boldsymbol{H}_e(N_0) \\ \hat{\boldsymbol{\theta}}_e(N_0) = \boldsymbol{B}^{-1}(N_0)\boldsymbol{H}_e^{\mathrm{T}}(N_0)\boldsymbol{Z}(N_0) \\ \boldsymbol{P}(N_0) = \boldsymbol{B}^{-1}(N_0) \end{cases} . \tag{18}$$

where $\boldsymbol{H}_e(N_0)$ and $\boldsymbol{Z}(N_0)$ are the identification matrix and observation matrix accumulated from 0 to $N_0$ steps, respectively, and then the identification parameter $\hat{\boldsymbol{\theta}}_e(N_0)$ of the $N_0$th step is solved by $\hat{\boldsymbol{\theta}}_e(N_0) = \boldsymbol{B}^{-1}(N_0)\boldsymbol{H}_e^{\mathrm{T}}(N_0)\boldsymbol{Z}(N_0)$, which is used as the initial quantity of the recursive least squares algorithm.

In the process of aerodynamic identification, in order to improve the identification accuracy, it is often necessary to add a certain excitation to the control input link, so as to excite the characteristics of each state in the aerodynamic model. According to [37], a quadrature optimization multi-sine excitation signal is added to the control input:

$$u_{ms} = \sum_{i \in \{1,2,\cdots,L\}} A \sin\left(\frac{2\pi it}{T_a} + \phi_i\right), \tag{19}$$

where $A$ is the amplitude of the multi-sine excitation signal, $\phi_i$ the phase angle for the $i$th sinusoidal component, $L$ the total number of available harmonic frequencies, and $T_a$ the length of the excitation time period.

### 4.2.2. Identification-Based Adaptive Dynamic Inversion Inner-Loop Control Law

The rotational dynamics equation in the UAV mathematical model Equation (1) can be written in the following general form:

$$\dot{\boldsymbol{\omega}} = \boldsymbol{I}^{-1}(\boldsymbol{M}_o + \boldsymbol{M}_\delta), \tag{20}$$

where $\boldsymbol{I}$ is the inertia matrix. $\boldsymbol{\omega} = \begin{bmatrix} p & q & r \end{bmatrix}^T$ consists of the three-channel angular rates, $\boldsymbol{M}_o = \begin{bmatrix} \overline{L}_o & \overline{M}_o & \overline{N}_o \end{bmatrix}^T$ consists of the external moments of the three axes about the center of mass, and $\boldsymbol{M}_\delta = \begin{bmatrix} \overline{L}_{\delta_a} & \overline{M}_{\delta_e} & \overline{N}_{\delta_r} \end{bmatrix}^T$ consists of the external moments of the three axes due to the deflection of the control surfaces.

Then the inner-loop control law can be constructed based on the proportional virtual control quantity of the control error as follows:

$$\boldsymbol{M}_{\delta,c} = \boldsymbol{I}\boldsymbol{K}_\omega(\boldsymbol{\omega}_c - \boldsymbol{\omega}) - \hat{\boldsymbol{M}}_o, \tag{21}$$

where $\boldsymbol{M}_{\delta,c}$ is the control moment command, which contains the estimate of the current moment on the UAV $\hat{\boldsymbol{M}}_o$ and is determined by the current identified model of the UAV.

The system dynamics are written in a form proportional to the desired behavior and everything else is set as a lumped input disturbance $\boldsymbol{\varepsilon}$ [38]:

$$\dot{\boldsymbol{\omega}} = \boldsymbol{K}_\omega(\boldsymbol{\omega}_c - \boldsymbol{\omega}) + \boldsymbol{u}_{adp} + \boldsymbol{\varepsilon}, \tag{22}$$

where $\boldsymbol{u}_{adp}$ is the proportional adaptation term of the control input, and $\boldsymbol{u}_{adp} = \boldsymbol{I}^{-1}\Delta\boldsymbol{M}$.

The angular rate observation error is defined as $\widetilde{\boldsymbol{\omega}} = \hat{\boldsymbol{\omega}} - \boldsymbol{\omega}$, and used to approximate $\boldsymbol{\varepsilon}$, so as to obtain the angular rate observation dynamic equation:

$$\dot{\hat{\boldsymbol{\omega}}} = \boldsymbol{K}_\omega(\boldsymbol{\omega}_c - \boldsymbol{\omega}) + \boldsymbol{u}_{adp} - \boldsymbol{K}_{adp}\widetilde{\boldsymbol{\omega}}. \tag{23}$$

Computing the error dynamics from Equations (22) and (23):

$$\dot{\widetilde{\boldsymbol{\omega}}} = -\boldsymbol{K}_{adp}\widetilde{\boldsymbol{\omega}} - \boldsymbol{\varepsilon}. \tag{24}$$

The Laplace transform of the error dynamics model Equation (24) shows that the adaptive control input $u_{adp}$ can filter out the low-pass part of the disturbance $\varepsilon$ by letting:

$$u_{adp}(s) \triangleq K_{adp}\widetilde{\omega}(s) = -\frac{K_{adp}}{s + K_{adp}}\varepsilon(s). \tag{25}$$

Finally, the adaptive term $u_{adp}$ is added to the inner-loop control law in Equation (21) to complete the design of the adaptive dynamic inversion inner-loop control law:

$$M_{\delta,c} = IK_{\omega}(\omega_c - \omega) - \hat{M}_o + IK_{adp}\widetilde{\omega}. \tag{26}$$

## 5. Simulation Result and Discussion

This section sets up two simulation cases in order to verify the robustness of the sliding mode disturbance observer-based adaptive dynamic inversion control law for composite disturbances and fault tolerance to structural failure of the control surface.

One case simulates the structural failure of the control surface, and verifies the fault tolerance of the adaptive dynamic inversion control method through mathematical simulation. The other case simulates the external composite disturbance, and verifies the improvement in the robustness of the sliding mode disturbance observer to the conventional dynamic inversion control method through mathematical simulation.

### 5.1. Fault Tolerance Verification of the Adaptive Dynamic Inversion Control Method

The pitch channel control simulation is taken as an example and the simulation conditions are set. The real-time aerodynamic identification-based adaptive dynamic inversion control simulation is divided into two stages. In stage 1, the control surface excitation signal is added and the real-time identification switch is turned on in the state of level flight at 7.5 km. In stage 2, after the level flight identification is completed, the pitch angle command is tracked, and the real-time identification calculation is still performed during this process.

The initial data of states, and the controller parameters, are set as shown in Table 1.

**Table 1.** Variable initialization and parameter settings.

| Description | Symbol | Value |
|---|---|---|
| flight altitude | $H$ | 7.5 km |
| flight velocity | $V$ | 150 m·s$^{-1}$ |
| angle of attack | $\alpha$ | 7.2° |
| NDI inner-loop bandwidth | $K_q$ | 12 |
| NDI inner-loop adaptive parameter | $K_{adp}$ | 0.05 |
| NDI outer-loop bandwidth | $\omega_{\theta d}$ | 4 |

The continuous $-50\%$ structural failure deviation of the elevator efficiency was injected at the time of 25 s, as shown by the red arrow in Figure 4a, and the conventional dynamic inversion and the real-time identification-based adaptive dynamic inversion were used for control, respectively. The simulation results are shown in the following figures.

Figure 4a shows that, in level flight, the two control methods have similar capabilities. The adaptive NDI control method is able to respond faster to commands while reducing control overshoot, which is reduced from 0.83° to 0.05°, when maneuvering occurs. In addition, when the partial loss damage of control surface occurs at 25 s, the conventional NDI control deviates significantly, and then gradually converges to the desired value, while the adaptive NDI control has always maintains high-precision attitude tracking, realizing fault-tolerant control of the control surface. The attitude tracking error curve is shown in Figure 4b. Figure 4c indicates that the angle of attack has a smaller fluctuation range under the control of the adaptive NDI compared to the conventional NDI control.

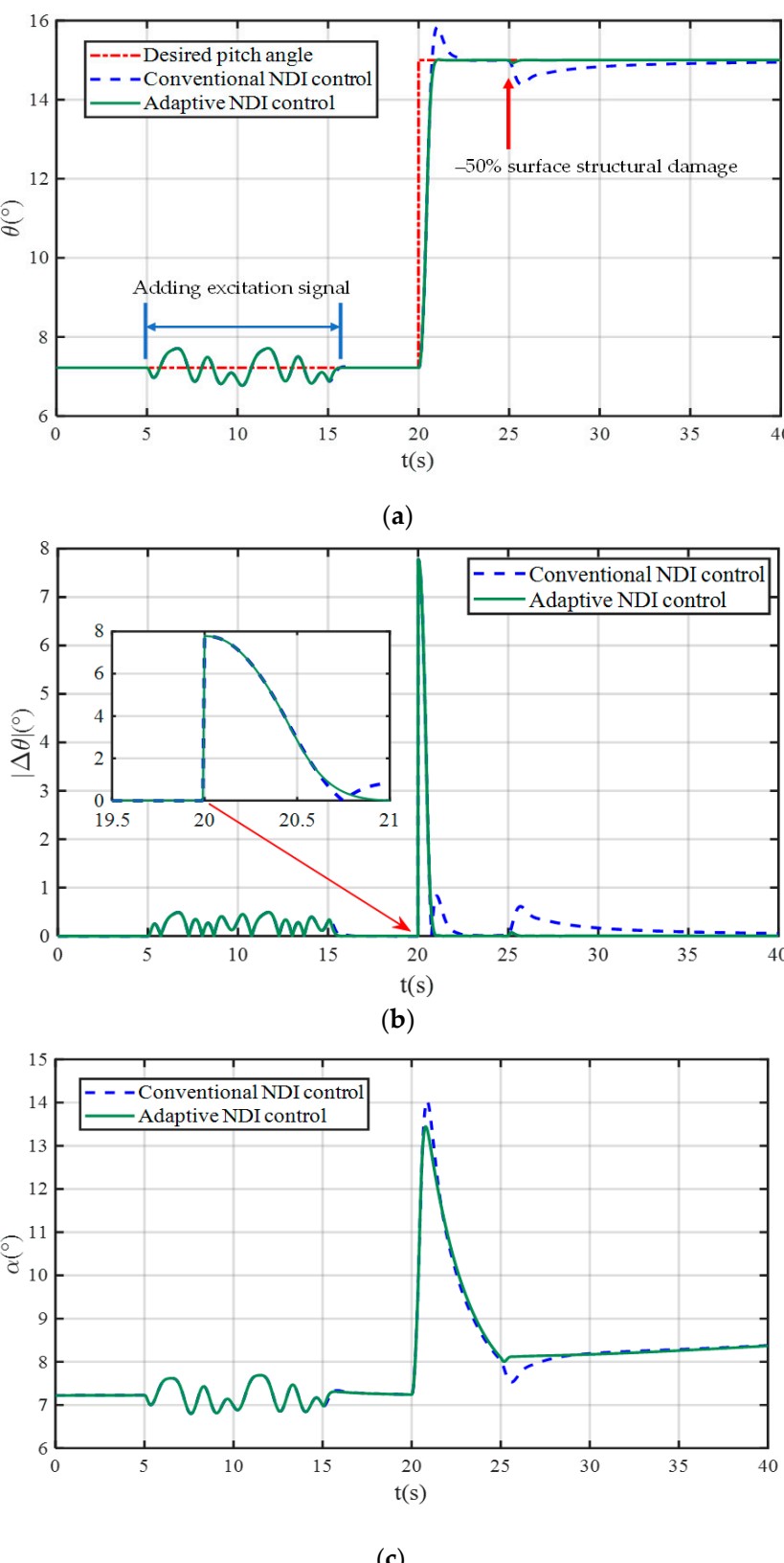

**Figure 4.** Simulation comparison of angle control under elevator surface structural failure. (**a**) The illustration of the difference between adaptive NDI and conventional NDI in command tracking under the condition of elevator failure. (**b**) Pitch angle tracking error under the two control methods. (**c**) The illustration of the change in the angle of attack when adaptive NDI and conventional NDI are controlled separately.

Figure 5 presents the identification result of elevator efficiency $C_m^{\delta_e}$ during the attitude angle tracking. In the control process, the identification was not implemented in the first 10 s, but the state data was accumulated. At 10 s, the initial value of elevator efficiency was obtained by the batch least squares method, and then the recursive least squares method was used to estimate the elevator efficiency in real time.

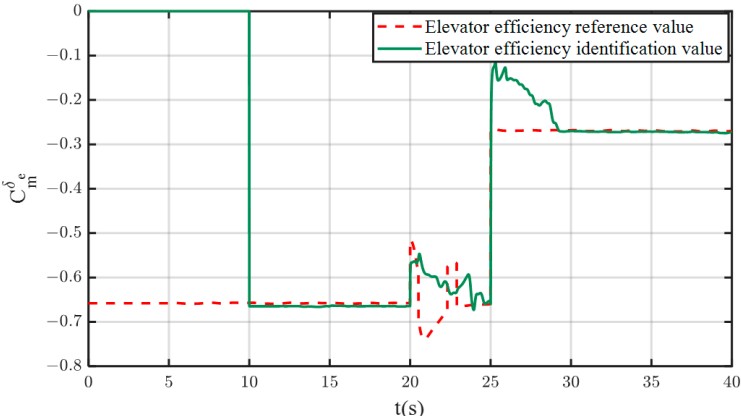

**Figure 5.** Simulation result of elevator efficiency identification.

The dynamic curves of the rudder surface and angular rate during the above maneuver are shown in Figure 6.

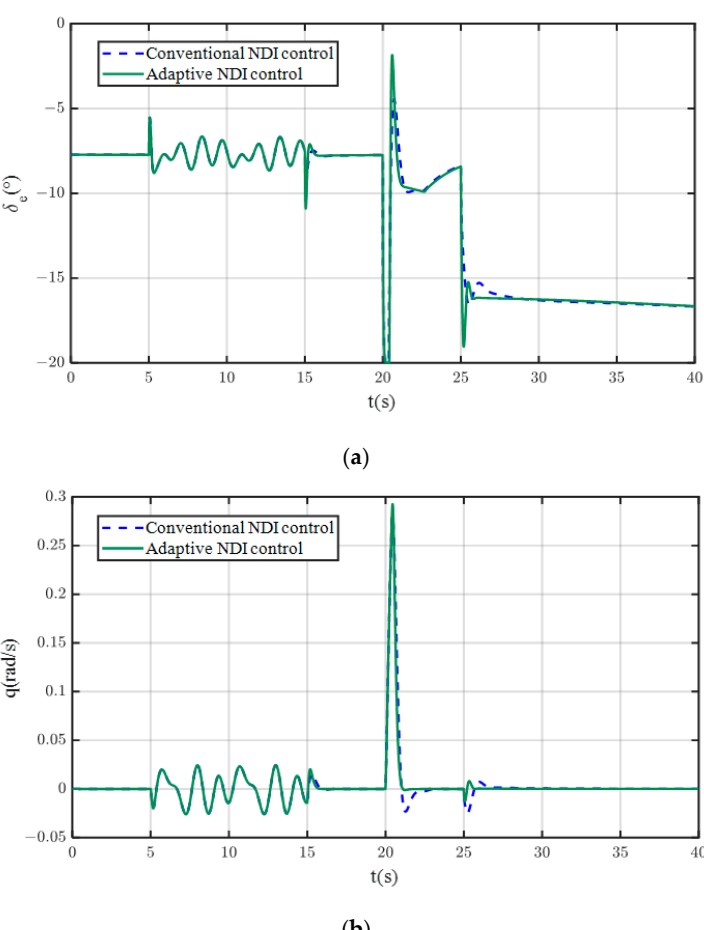

(**a**)

(**b**)

**Figure 6.** Comparison of elevator simulation and angular rate simulation under conventional NDI and adaptive NDI control. (**a**) Elevator deflection curves under conventional NDI and adaptive NDI control. (**b**) Angular rate variation curves under conventional NDI and adaptive NDI control.

Figure 6a shows that under adaptive NDI control, the elevator is deflected more violently during maneuvers and failures compared to conventional NDI. However, the inertial response link of the actuator is added to the simulation results, so in order to achieve more precise attitude control, this deflection of the elevator is acceptable. Figure 6b illustrates that under the control of adaptive NDI, the angular rate is non-zero atvery few times; that is, the attitude control is more stable.

### 5.2. Disturbance Suppression Verification of the ADSIC Method

When the UAV encounters severe composite external disturbances, a single real-time aerodynamic identification-based adaptive dynamic inversion control method often cannot suppress it; thus, the outer-loop sliding mode disturbance observer is introduced. At this time, the controller includes the baseline dynamic inversion, the outer-loop sliding mode disturbance observer, and the inner-loop adaptive aerodynamic identification module, which are collectively called the adaptive disturbance suppression integrated controller (ADSIC). Here, the ADSIC is verified by numerical simulation.

The simulation initial conditions are the same as those in Section 5.1, and the controller parameters are set as shown in Table 2.

**Table 2.** ADSIC control parameters settings.

| Description | Symbol | Value |
|---|---|---|
| sliding mode observer constant coefficients | $w_{\theta 1}, w_{\theta 2}$ | 1.9, 0.02 |
| NDI inner-loop bandwidth | $K_q$ | 12 |
| NDI inner-loop adaptive parameter | $K_{adp}$ | 0.05 |
| NDI outer-loop bandwidth | $\omega_{\theta d}$ | 4 |

From the above table, except for the newly added sliding mode interference observer, the other relevant parameters of the dynamic inversion are the same as before.

The theoretical composite disturbance $D_{\theta s}$ of the pitch angle loop is defined as a sinusoidal signal, as shown in Equation (27). Then, according to the theoretical analysis in Section 4.1, the sliding mode observer estimates $D_{\theta s}$, namely $\hat{D}_{\theta s}$, so as to compensate the control law and realize the suppression of the disturbance.

$$D_{\theta s} = 0.0873 \sin 1.5t. \tag{27}$$

The initial state was considered to be the same as that of the previous simulation, and the adaptive dynamic inversion control law based on online aerodynamic identification was kept in the working state. A $-50\%$ elevator surface missing failure also occurred at 25 s. In addition, at the moment of 30 s, the sinusoidal composite interference signal was injected, and then the control simulation comparison between ADSIC and adaptive NDI was carried out, as shown in Figure 7.

It can be seen from Figure 7a that the adaptive NDI cannot suppress the deviation in the attitude angle in the case of strong external disturbance, while the ADSIC control integrated with the adaptive aerodynamic identification module and the sliding mode observer can achieve fault-tolerant control, and can also effectively reduce the large vibration of the controlled pitch angle caused by external composite disturbance. The attitude tracking error curve is shown in Figure 7b.

Then, the elevator deflection curve, the angular rate variation curve, and the observation curve of the sliding mode observer to the composite disturbance in the above control process are given in Figure 8.

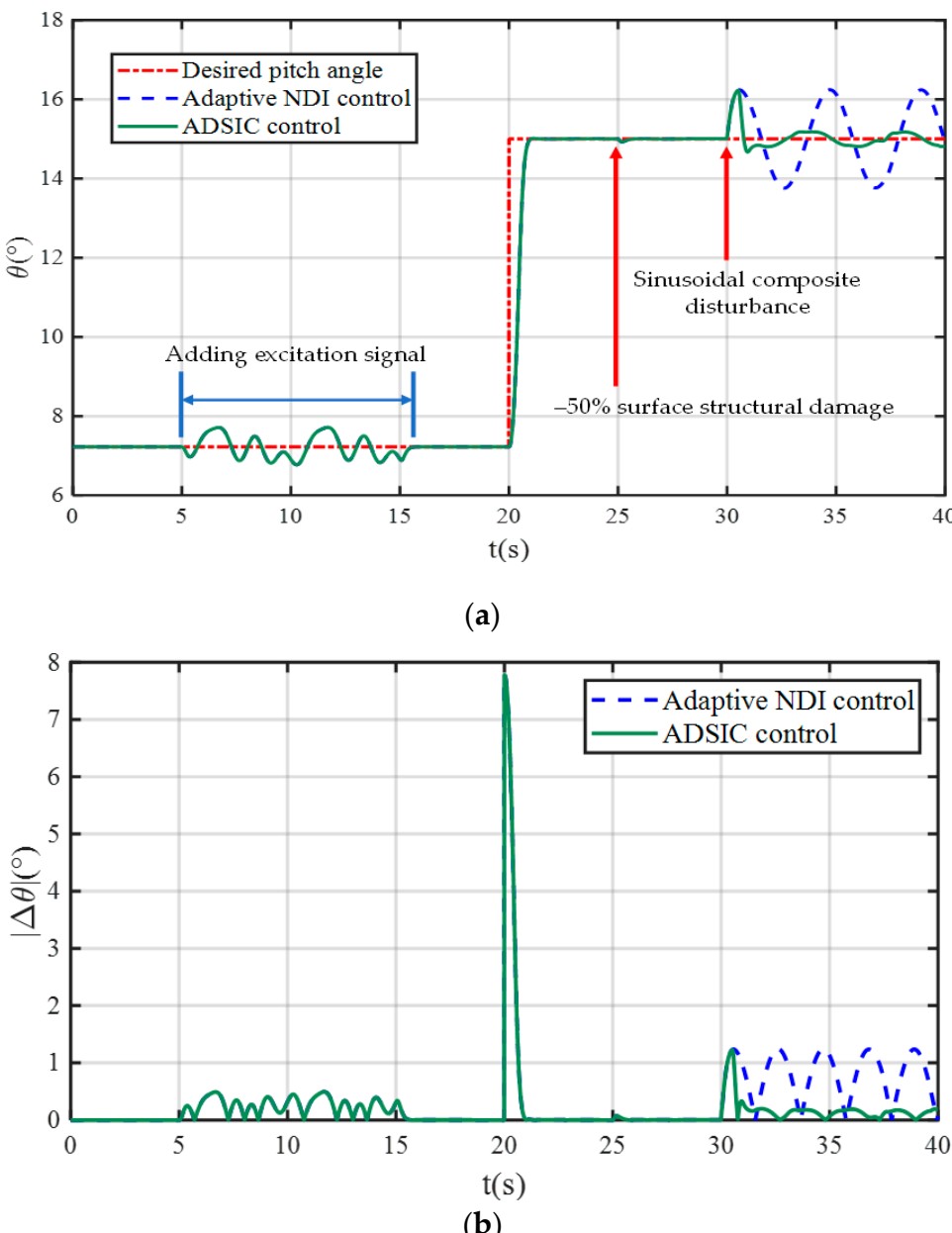

**Figure 7.** Simulation comparison of pitch angle control under composite disturbance. (**a**) The illustration of the difference between adaptive NDI and ADSIC under composite disturbance. (**b**) Pitch angle tracking error under the two control methods.

Figure 8a shows that, due to the control compensation of the sliding mode observer, the elevator oscillates significantly in the early stage of the composite disturbance. It is acceptable for high-precision attitude control, and a first-order system reflecting the dynamic characteristics of the elevator actuator was added in the elevator simulation curve in Figure 8a. Figure 8b shows that, at the initial moment of introducing the sliding mode observer, the angular rate has a peak, and then quickly stabilizes. Compared with the ANDI control, the angular rate under the ADSIC control has a certain phase lead. Figure 8c shows the estimation of the composite disturbance by the sliding mode observer. It can be seen from the figure that the sliding mode disturbance observer does not work before 30 s. After the disturbance occurs at 30 s, its estimation by the sliding mode observer has an initial fluctuation, and then immediately converges to the true value of the disturbance, with an accuracy of more than 98%.

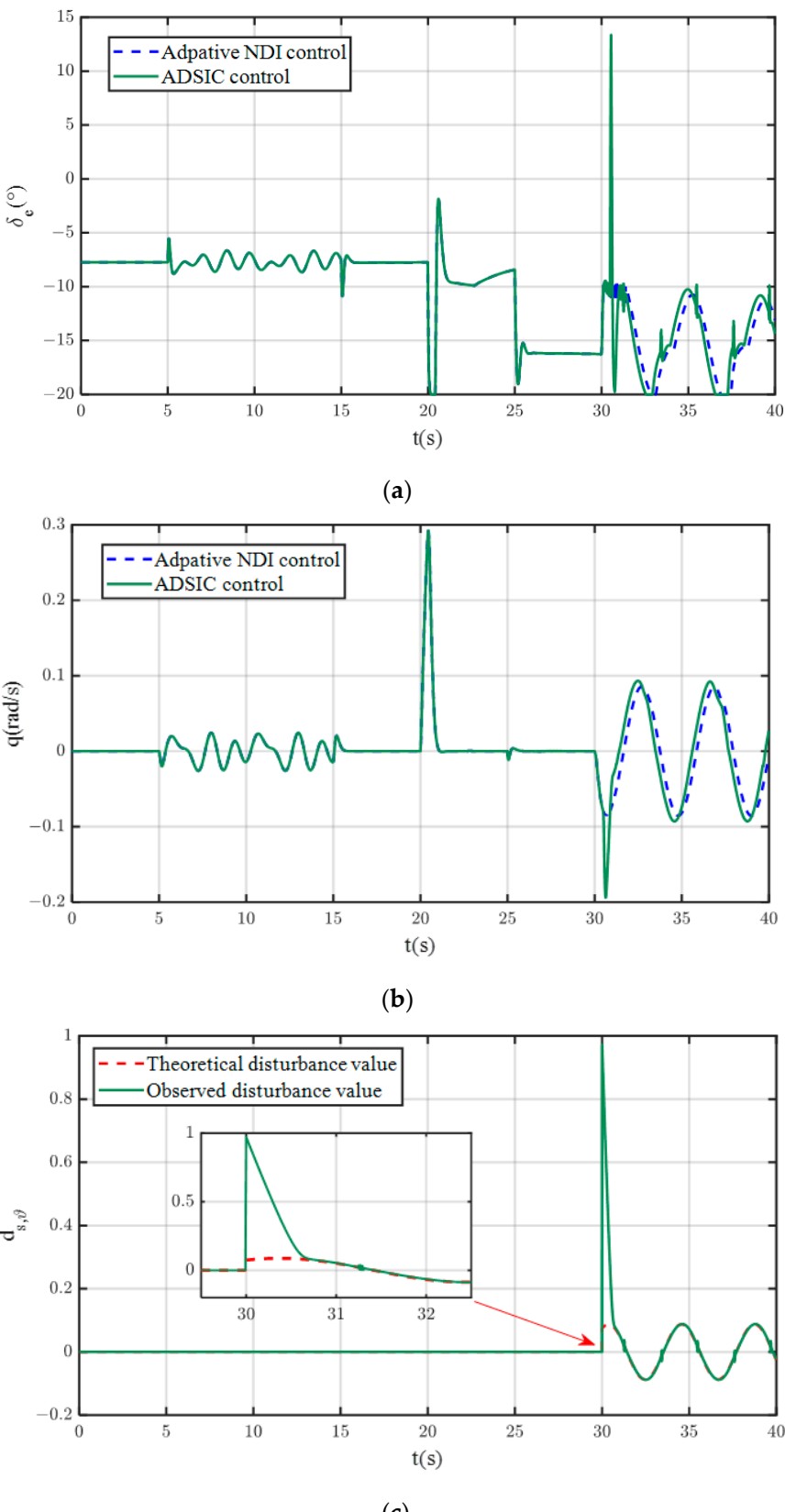

**Figure 8.** Simulation of key variables in attitude control. (**a**) Elevator deflection curves under adaptive nonlinear dynamic inversion (ANDI) and ADSIC control. (**b**) Angular rate variation curves under ANDI and ADSIC control. (**c**) Simulation of sliding mode observer's estimation of compound disturbance.

## 6. Conclusions

In this study, a sliding-mode disturbance observer-based adaptive dynamic inversion fault-tolerant controller is designed to address the problem of poor fault tolerance and anti-disturbance capability of UAVs under conventional control. First, the UAV attitude angle dynamic equations and the rotational dynamic equations with partial loss of actuator surface are established. Then, the adaptive fault-tolerant control law is designed under the framework of nonlinear dynamic inversion control. A super-twisting algorithm-based sliding mode disturbance observer is introduced into the dynamic inversion outer loop to estimate the composite disturbance, thereby compensating the outer loop control law. In addition, an online aerodynamic identification module is introduced into the dynamic inversion inner loop to update the aerodynamic model of the UAV, thereby realizing fault-tolerant control for the partial loss of control surfaces. The above two aspects constitute the adaptive fault-tolerant controller. Numerical simulation verification results indicate that the proposed control method can avoid attitude control deviation in the event of structural failure of the control surface. When the UAV encounters composite disturbance, the method can quickly suppress the fluctuation in attitude, showing strong robustness and adaptive ability.

Furthermore, multi-actuator redundancy is one of the characteristics of the advanced UAV, providing a physical guarantee for fault-tolerant control. In this regard, aerodynamic identification-based adaptive control allocation appears to be a promising approach to achieve reconfigured control of UAVs under different failure scenarios, which will be the direction of further research.

**Author Contributions:** Conceptualization, Z.D. and K.L.; methodology, Z.D.; software, Z.D.; validation, Z.D.; formal analysis, Z.D. and K.L. and S.W.; investigation, Z.D. and S.W.; resources, Z.D. and K.L.; data curation, Z.D. and K.L.; writing—original draft preparation, Z.D.; writing—review and editing, Z.D. and K.L. and S.W.; visualization, Z.D.; supervision, K.L.; project administration, K.L.; funding acquisition, K.L. All authors have read and agreed to the published version of the manuscript.

**Funding:** This research was funded by National Natural Science Foundation of China, grant number U2141229 and Aeronautical Science Foundation of China, grant number 2019ZC063001.

**Institutional Review Board Statement:** Not applicable.

**Informed Consent Statement:** Not applicable.

**Data Availability Statement:** Not applicable.

**Conflicts of Interest:** The authors declare no conflict of interest.

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
