# Peer review of "Sliding Mode Disturbance Observer-Based Adaptive Dynamic Inversion Fault-Tolerant Control for Fixed-Wing UAV"

_drones, doi:10.3390/drones6100295_

Round 1
Reviewer 1 Report
Manuscript ID: drones-1932280
Review Report
Dear Editors and Authors,
The reviewer would like to submit the reviewed manuscript for the study entitled "Sliding Mode Disturbance Observer-Based Adaptive Dynamic Inversion Fault Tolerant Control for Fixed-Wing UAV" under consideration for publication in Drones (ISSN 2504-446X).
After the review process, the reviewer would like to give some critical thinking and idea to help authors get their job done efficiently.
In this study, the authors proposed sliding mode disturbance observer-based adaptive dynamic inversion fault tolerant control, which is attractive to readers. UAVs are frequently used in the military. Due to flying unpredictability and malfunctions, the fixed-wing UAV's control system must be well-designed. This work builds an outer loop sliding mode disturbance observer-based disturbance suppression dynamic inversion controller and an inner loop real-time aerodynamic identification-based adaptive fault-tolerant active inversion controller. The external loop controller's sliding mode disturbance observer uses second-order super-twisting to reduce chatter. The internal loop controller's aerodynamic identification uses recursive least squares to update the UAV's aerodynamic model online, enabling fault-tolerant control for damaged control surfaces. Mathematical simulation validates the proposed SMDO fault-tolerant adaptive control approach. In conclusion, the reviewer found that the paper has merits and could be acceptable to publish in future forms. Therefore, please revise the manuscript according to the reviewer’s comments.
- Please add some keywords to help readers find your work. Usually, each scientific article contains 5 to 10 keywords that are not repeated in words/phrases in the title.
- Figures 1 and 2: Please explain these figures in the text. The reviewer did not see the author presenting these 2 Figures.
- Please explain or mention equations (2, 3, 13, 15, 18,19, 20,25, 26, 27) in the text.
- Lines 47, 50, 56" remove the word "reference" before the citations [11,12,14]
- Figure 3: Please consider “Is convergence?” instead of “Whether to Converge”
- Let's add Eq. before the equations you want to mention in the texts. For example, Eq. (4), Eqs. (8-10) in lines 227-228, as you remarked in Eq. (11) in line 197 or Eq. (23), Eq. (24) in Line 288.
- Line 317, 379: which figure are you mentioning? Please revise.
- Line 312: Table 1: please consider using the unit (m.s-1) instead of m/s
- The literature study is quite exhaustive and contains a lot of supporting evidence.
- Your concept is a novel contribution to the ongoing research.
- In conclusion, it is suggested to summarize the whole process of this study (step by step) again. The authors should show their contributions to this study. Besides, please present the limitation of the study and their further studies.
The reviewer does not detect any significant changes. The reviewer hopes that his point of view could help the authors improve their work well.
Sincerely yours,
The Reviewer
Reviewer 2 Report
The authors present a sliding mode disturbance observer-based (SMDO) adaptive dynamic inversion fault tolerant controller. The method is helpful for the fault-tolerant control for the control surface damage. There are some comments as follows.
1. The introduction should emphasize the importance and necessity of the fault-tolerant control.
2. Why the influence of aerodynamic model changes on control after UAV failure is needed to be considered?
3. What is the specific form of the control surface damage?
4. What are the advantages of the method mentioned in this paper compared to existing control methods?
5. More information should be given in the result part, such as the errors of the pitch angles and the root mean square of errors, to prove the effectiveness of control.
6. In Fig.5(a) and Fig.7(a), the elevator response fluctuates sharply. Will this cause rate saturation of control surface? How does the author consider this problem?
7. What are the limitations of this sliding-mode disturbance observer-based adaptive dynamic inversion fault-tolerant controller? And what are the drawbacks and difficulties of the proposed method implementation in real world applications?
Reviewer 3 Report
This article proposes a sliding-mode disturbance observer-based adaptive dynamic inversion fault-tolerant controller is designed to address the problem of poor fault tolerance and anti-disturbance capability of UAV under conventional control. The obtained results seem meaningful, but the paper cannot be published in current version. There are several suggestions:
1.The real-time aerodynamic model identification is added in the control structure, but its effect cannot be found in the simulation test.
2. Since the aerodynamic mode can be identified in real-time, which means the control law can be refreshed in real-time. So, is an adaptive controller necessary under this condition?
3. The authors try to combine the strengths of many approaches, i.e., from the title. However, these approaches need to be a trade-off under some circumstances, and the necessity of each module in ADSIC is not stated clearly.
4. The simulation looks good, but some pictures are not clear enough, such as Figure 7(c), where is the red line?
5. The references are relatively out-of-date, some new papers within three years should be cited.
6. There are some grammatical errors, and the authors should tackle the problem of English tense.
Round 2
Reviewer 3 Report
No more comments